# Slotting blasting model experiment and PCA-PNN evaluation model of influencing factors of slitting effect

Qiang Li[1]☺, Jinshan Sun[1]☺*, Xianqi Xie[1]☺, Qian Dong[1]‡, Jianguo Wang[2]‡, Nan Jiang[1]‡

**1** State Key Laboratory of Precision Blasting Engineering, Jianghan University, Wuhan, Hubei, PR China,
**2** Faculty of land Resources Engineering, Kunming University of Science and Technology, Kunming, Yunnan, PR China

☺ These authors contributed equally to this work.
‡ These authors also contributed equally to this work.
* sun99001@126.com

## Abstract

Numerous parameters influence the slotting performance of slotted cartridge, to facilitate rapid, efficient, and accurate predictions of the slitting performance, statistical analysis of PMMA blasting experiments with six different slitted cartridge parameters yielded 12 evaluation indicators. Subsequently, a principal component analysis (PCA) method was introduced to reduce the dimensionality of the data associated with these indicators, and three new comprehensive indicators were extracted for a comprehensive assessment of the slotting performance. The PCA scores ranked the influence of the six slotted cartridge parameters on slotting performance as follows: decoupling coefficient, slotting width, slotting angle, slotting tube thickness, slotting tube material, and charge amount. This ranking serves as a guideline for selecting suitable slotted cartridge parameters. The predictive results demonstrated that the PCA-PNN model performed well across eight different training and testing sample configurations, achieving correct prediction rates of 100%, 100%, 96.43%, 96.43%, 92.86%, 89.29%, 89.29% and 85.71%, respectively. This corresponded to an average accuracy improvement of 12.95% compared to data that were not subjected to PCA dimensionality reduction. Moreover, the PCA-PNN model was validated as a robust and feasible approach for evaluating the slotting performance of slotted cartridge.

## Introduction

Slotted cartridges are designed with slots of various angles, shapes, and quantities on the slotting tube. These slots control the distribution of stress and the movement direction of explosive gases, thereby controlling the initiation and propagation direction of cracks and reducing the disturbance to the rock mass caused by blasting [1,2].

**Data availability statement:** All relevant data are within the manuscript.

**Funding:** The National Key Research and Development Program (2021-008), Wuhan key Research and Development Program (2024050802030155), 2024 Chutian Talent Plan—Science and Technology Innovation Team Project. The funders had no role in study design, data collection and analysis, decision to publish, or preparation of the manuscript.

**Competing interests:** The author(s) declared no potential conflicts of interest with respect to the research, authorship, and/or publication of this article.

Many scholars [3,4] have demonstrated through numerical simulations, model experiments, and field tests that slotted cartridges are effective in reducing damage to rock mass and forming a well-defined contour. Current research on individual parameters of slotted cartridges mainly focuses on slotting width [5], slotting angle, slotting tube material [6,7], decoupling coefficient [8,9], slotting tube thickness, and charge amount [10,11]. Kang et al. [12] studied the formation mechanism and propagation characteristics of explosive cracks of slit charges with different slit numbers. Xie et al. [13] studied the crack propagation of slotted cartridges at different micro-delay times through similar simulation experiments. Man et al. [14] investigated the slotted cartridge blasting in high-level radioactive waste disposal projects using dynamic mechanical parameters. Yang et al. [15] demonstrated the superiority of slotted cartridge by optimizing blasting parameters and performing full-section controlled blasting. Numerous other scholars have extensively studied slotted cartridges under different confining pressures [16,17], varying joint and fracture conditions, double-hole blasting [18,19], and other diverse conditions [20]. While research on slotted cartridges is well-developed, systematic sensitivity analysis of various parameters on the slitting effect is lacking. There are numerous indicators for evaluating the slotting performance of slotted cartridge, and identifying effective and appropriate indicators has become a pressing issue that needs to be addressed.

Therefore, for the multiple parameters of slotted cartridges, it is necessary to perform dimensionality reduction of relevant data, feature extraction, and probability classification. At present, there are few such algorithm models used in the field of blasting. Nitin et al. [21] proposed a method to predict porosity from seismic inversion data by using Principal Component Analysis (PCA) and Probabilistic Neural Network (PNN) techniques. Wu et al. [22] used PCA-PNN to predict the rockburst intensity and proved the rationality of the model. Wang et al. [23] evaluated the risk level of rock burst in coal mining through R-type factor analysis and PNN model. Yu et al. [24] aimed at the demagnetization fault problem of the permanent magnet synchronous motor (PMSM), a demagnetization fault diagnosis method based on the combination of the principal component analysis (PCA) algorithm, the improved sparrow search algorithm (ISSA), and the probabilistic neural network (PNN) algorithm is proposed. Ahmadi et al. [25] identifies the same optimal configuration through the artificial intelligence (AI) -driven optimization via a hybrid Convolutional Neural Networks and Genetic Algorithms. Ahmadi et al. [26] used the Grey Wolf Optimization to optimize the geometric structure of the vortex generator, which improved the thermal performance by 36.8%, while the pressure drop increased by only 18.3%. Hong et al. [27] used Particle Swarm Optimization (PSO) to improve the parameter selection process of Probabilistic Neural Network (PNN) model to evaluate the disaster-bearing level of each unit. Mohamad et al. [28–30] studied the bonding strength and bonding quality between different metal laminates through the PNN model. The combined use of algorithms [31,32] is widely used in various projects.

Currently, the use of slotted cartridge is primarily determined by comparative tests or previous experience. However, this method is only applicable to field construction with fixed bore diameter, fixed materials, and fixed slotting width. Under other

conditions, the parameters of slotted cartridge cannot be effectively determined, and previous methods fail to meet engineering requirements. To date, evaluation of slotting performance levels of slotted cartridge remains a research challenge. Therefore, a method to distinguish the slotting performance levels of slotted cartridge should be sought. In view of this, we conducted experiments on various influencing parameters of slotted cartridge and collected as many evaluation indicators as possible to gain a comprehensive understanding of the slotting performance level. However, the numerous evaluation indicators have become an obstacle to solving the problem. The slotting effect is classified by the PNN network, but when the Gaussian function is used as the activation function in a PNN network, the indicator variables must be uncorrelated and identically distributed [33]. Since there is a certain correlation between the slotting performance evaluation indicators for slotted cartridge, it is necessary to eliminate this correlation before applying the PNN network. Based on this, an evaluation model for slotting performance level of slotted cartridge that combines the PCA and PNN was proposed. The PCA method was used to compress and extract characteristic information from the evaluation indicator data, i.e., a small number of variables were utilized to reflect as much information as possible about the variables of original slotted cartridge's slotting performance to ensure minimal loss of original information and a minimal number of variables. The principal components after dimensionality reduction were used to rank the factors influencing the slotting performance of slotted cartridge according to the PCA scores and to obtain the ranking of various factors' influence on slotting performance [22]. A PCA-PNN model was established based on the PNN for the evaluation of slotting performance level of slotted cartridge, and its feasibility and effectiveness were validated through PMMA model experiments.

## Blasting experiments with different slotted cartridge parameters

### Experimental scheme

There are many factors that affect the slotting performance of slotted cartridge. Here, the decoupling coefficient $k$, slotting width $d$, slotting tube thickness $l$, slotting tube material $t$, charge amount $p$, and slotting angle $a$ were selected for study. The experimental scheme is detailed in Table 1, and some schematic diagrams of slotting tubes are displayed in Fig 1(a). (1) Stick the slit of the slotting tube with tape (prevent lead azide leakage), put the slotting tube into the center of the blast hole and fill the bottom with rubber mud, and the holes on the PMMA are sealed with tape, as shown in Fig 1(b). (2) The corresponding amount of lead azide was added to the slotting tube, a self-made detonation probe (Fig 1(c)) was inserted. Then the rubber mud is used to seal the orifice (Fig 1(d)), and the top was glued with tape to carry out the experiment. (3) Fixtures are used on both sides of the blast hole to clamp the blasthole, so as to prevent the blasting energy from escaping to both sides of the blasthole. (4) Connect the initiation line on the initiation probe to the initiator (Fig 1(e)), and complete the initiation after charging.

### Data processing

(a) Area of crushed zone

The radius of the crushed zone was selected based on the photographs, and the number of black pixels in this zone was programmatically calculated. The area of the crushed zone was calculated according to a pixel size of 400 pixels/cm for photographs.

(b) Damage variable

A damage variable $\omega$ was defined, where $\omega = 0$ indicates no damage and $\omega = 1$ indicates complete failure. Since the experimental method employed in this paper only analyzes the macroscopic cracks produced by blasting, $\omega$ characterizes the damage and failure of the material due to macroscopic cracks and reflects the extent of material damage [34]. The $\omega$ was defined by Eq. (13). MATLAB code was written to calculate the damage based on Eq. (13).

**Table 1. Model experimental scheme.**

| number | parameter | charge diameter/mm | blasthole diameter/mm | slotting tube thickness/mm | slotting tube material | slotting width/mm | charge amount/mg | slotting angle/° |
|---|---|---|---|---|---|---|---|---|
| 1 | k | 6 | 9 | 1 | PVC | 0.5 | 100 | 180 |
| 2 | | 6 | 10 | 1 | PVC | 0.5 | 100 | 180 |
| 3 | | 6 | 11 | 1 | PVC | 0.5 | 100 | 180 |
| 4 | | 6 | 12 | 1 | PVC | 0.5 | 100 | 180 |
| 5 | | 6 | 13 | 1 | PVC | 0.5 | 100 | 180 |
| 6 | | 6 | 14 | 1 | PVC | 0.5 | 100 | 180 |
| 7 | | 6 | 15 | 1 | PVC | 0.5 | 100 | 180 |
| l | l | 6 | 10 | 0.5 | PVC | 0.5 | 100 | 180 |
| 9 | | 6 | 10 | 1 | PVC | 0.5 | 100 | 180 |
| 10 | | 6 | 10 | 1.5 | PVC | 0.5 | 100 | 180 |
| 11 | t | 6 | 10 | 1 | PVC | 0.5 | 100 | 180 |
| 12 | | 6 | 10 | 1 | Al | 0.5 | 100 | 180 |
| 13 | | 6 | 10 | 1 | Fe | 0.5 | 100 | 180 |
| 14 | d | 6 | 10 | 1 | PVC | 0.5 | 100 | 180 |
| 15 | | 6 | 10 | 1 | PVC | 0.8 | 100 | 180 |
| 16 | | 6 | 10 | 1 | PVC | 1.0 | 100 | 180 |
| 17 | | 6 | 10 | 1 | PVC | 1.5 | 100 | 180 |
| 18 | | 6 | 10 | 1 | PVC | 2.0 | 100 | 180 |
| 19 | p | 6 | 10 | 1 | PVC | 0.5 | 110 | 180 |
| 20 | | 6 | 10 | 1 | PVC | 0.5 | 120 | 180 |
| 21 | | 6 | 10 | 1 | PVC | 0.5 | 130 | 180 |
| 22 | a | 6 | 10 | 1 | PVC | 0.5 | 100 | 90 |
| 23 | | 6 | 10 | 1 | PVC | 0.5 | 100 | 105 |
| 24 | | 6 | 10 | 1 | PVC | 0.5 | 100 | 120 |
| 25 | | 6 | 10 | 1 | PVC | 0.5 | 100 | 135 |
| 26 | | 6 | 10 | 1 | PVC | 0.5 | 100 | 150 |
| 27 | | 6 | 10 | 1 | PVC | 0.5 | 100 | 165 |
| 28 | | 6 | 10 | 1 | PVC | 0.5 | 100 | 180 |

$$\omega = \frac{A_\omega}{A} = \frac{n_\omega}{n} \tag{1}$$

where $A_\omega$ is the damaged area of cracks in the image; $A$ is the total area of the image; $n_\omega$ is the number of pixels in the damaged area; and $n$ is the total number of pixels in the image.

(c) Fractal dimension

Currently, methods for calculating fractal dimension include similarity dimension, Hausdorff dimension, information dimension, box-counting dimension, correlation dimension, capacity dimension, and spectral dimension. Among these, the box-counting dimension visualizes the occupancy of the object in the study area and is relatively simple in mathematical computation. Therefore, it is widely used in fractal studies [35,36]. In this paper, the box-counting method was adopted to calculate the fractal dimension of each zone after blasting, which can be expressed by Eq. (14) [37]:

$$\lg N(\delta) = D \lg \delta + b \tag{2}$$

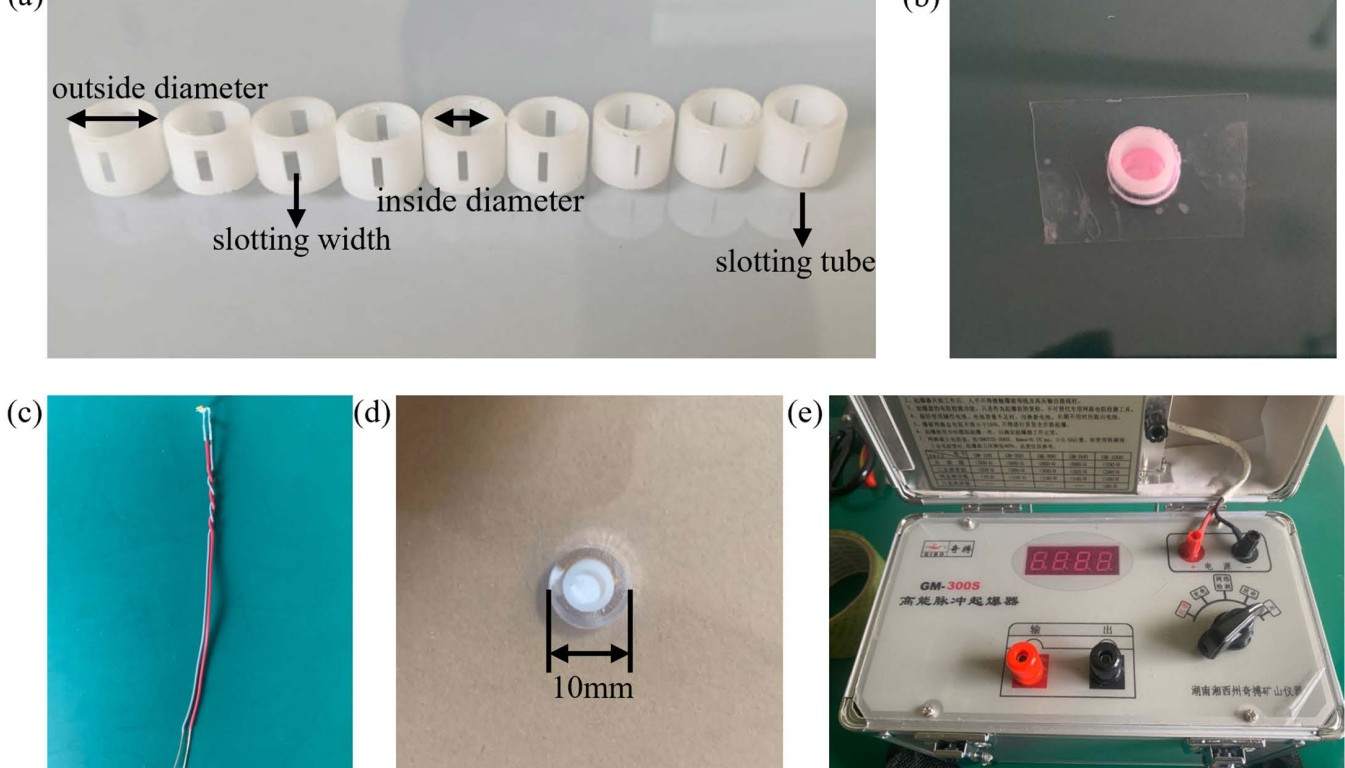

**Fig 1. (a) Schematic diagram of slotting tube; (b) Placement of slotting tube; (c) Detonation probe; (d) Charging; (e) High energy pulse initiator.**

In this study, the resolution of experimental images after blasting is 8000 pixels×8000 pixels, with a pixel size of 400 pixels/cm. Based on the principle of the box-counting method mentioned above, MATLAB programming was utilized to calculate the box-counting dimension of the binary images of blasting-induced cracks.

## Experimental results

Dynamic caustics experiments were conducted according to the above experimental scheme, and the results are detailed in Fig 2. By measuring 28 images in Fig 2 using CAD and writing MATLAB code for damage and fractal calculations, 12 indicators for evaluating the slotting performance of slotted cartridge, including average primary crack length(X1), average secondary crack length(X2), radius of crushed zone(X3), primary crack length/secondary crack length(X4), total number of cracks(X5), expected primary crack angle deviation(X6), number of main cracks(X7), area of crushed zone(X9), over-all damage variable(X9), slotting direction damage variable/non-slotting direction damage variable(X10), overall fractal dimension(X11), and slotting direction fractal dimension/non-slotting direction fractal dimension(X12), were obtained.

The experimental results are compared with the relevant scholars' research on the slitted cartridges. When the decoupling coefficient [9] is 1.67, the slitting effect is the best. The main crack length is positively correlated with the slotting tube thickness [38], but the crack length increases slowly with the increase of the tube thickness. The slotting tube material [7] has a great influence on the slitting effect. The effect of iron tube is the best, the effect of aluminum pipe is in the middle, and the effect of PVC pipe is the worst. For the slotting width [39], when the slotting

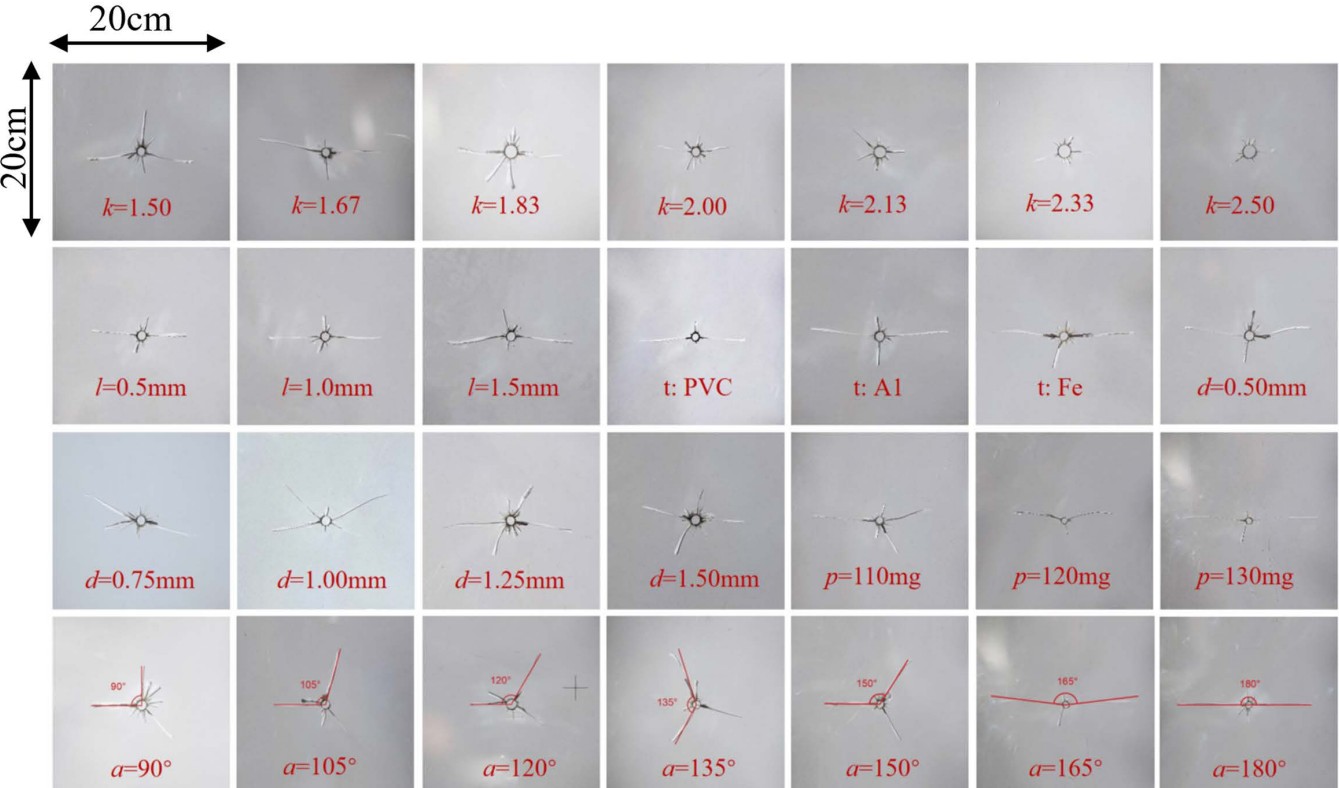

**Fig 2. Results of model experiment.**

width ≥ 0.75 mm, two main cracks with angle will be produced in the slotting direction. As the slotting width increases, the angle between the main cracks increases. The larger the charge amount [11], the longer the main crack; Different cutting angles [40], the crack can basically achieve the purpose of directional fracture; the longest crack decreases with the increase of the slotting angle, and the above conclusions are consistent with the related slotting cartridge research.

## PCA-based analysis

### Principal component analysis

The core of principal component analysis (PCA) is to transform the original indicators into a new set of independent, comprehensive indicators that contain most of the original information through linear combinations, thus achieving the dimensionality reduction [41–42]. By ensuring sufficient information in the original data, dimensionality reduction of the original multi-dimensional indicator variable data can compress the number of indicators, simplify the data, and reduce the time consumed in predicting the slotting performance levels of slotted cartridge. The model of this method is as follows:

$$X = \begin{bmatrix} x_{11} & x_{12} & \cdots & x_{1n} \\ x_{21} & x_{22} & \cdots & x_{2n} \\ \vdots & \vdots & \vdots & \vdots \\ x_{m1} & x_{m2} & \cdots & x_{mn} \end{bmatrix} \tag{3}$$

Assuming that there are $m$ samples and $n$ indicators, and that the value of the j-th indicator in the i-th sample is $x_{ij}$, the original data matrix is constructed as $X = (x_{ij})_{m \times n}$:

The original variable indicators are $x_1, x_2,..., x_n$. Let the comprehensive indicators after dimensionality reduction, i.e., the principal components, be $y_1, y_2,..., y_n$. The $n$ indicator vectors of X are used to make a linear combination, i.e.,:

$$\begin{cases} y_1 = l_{11}x_1 + l_{21}x_2 + \cdots + l_{n1}x_n \\ y_2 = l_{12}x_1 + l_{22}x_2 + \cdots + l_{n2}x_n \\ \vdots \\ y_1 = l_{1n}x_1 + l_{2n}x_2 + \cdots + l_{nn}x_n \end{cases} \tag{4}$$

Eqs. (1) and (2) satisfy that the sum of the squares of the coefficients of each equation is 1, as displayed in Eq. (3). Any two principal components are independent and uncorrelated. $y_1$ is the linear combination of $x_1, x_2,..., x_n$ with the maximum variance, $y_2$ is the linear combination of $x_1, x_2,..., x_n$ uncorrelated with $y_1$ with the maximum variance, and $y_n$ is the linear combination of $x_1, x_2,..., x_n$ uncorrelated with $y_1, y_2,..., y_{n-1}$ with the maximum variance.

$$l_{1j}^2 + l_{2j}^2 + \cdots + l_{nj}^2 = 1 \quad (j = 1, 2, \cdots, n) \tag{5}$$

To eliminate the incommensurability of indicators, it is necessary to perform dimensionless processing on the indicators before conducting PCA. The specific method is as follows:

$$x_{ij}^* = (x_{ij} - \overline{x_j}) / S_j \quad (i = 1, 2, \cdots, m; j = 1, 2, \cdots, n) \tag{6}$$

$$\overline{x}_j = \sum_{i=1}^{m} x_{ij}/m, \, s_j = \sqrt{\sum_{i=1}^{m} (x_{ij} - \overline{x}_j)^2 / (m-1)} \tag{7}$$

where $\bar{x}_j$ and $s_j$ are the mean and standard deviation of the j-th indicator in the i-th sample, respectively.

Calculate the Pearson correlation coefficient matrix between the indicators, i.e.,:

$$R = (r_{kl})_{n \times n} \quad (k, l = 1, 2, \cdots n) \tag{8}$$

where $r_{kl}$ is the correlation coefficient between the k-th and l-th indicators, and $r_{kl} = r_{lk}$ (i.e., a symmetric matrix). The calculation formula is:

$$r_{kl} = \frac{\sum\limits_{i=1}^{m} (x_{ik} - \overline{x_k})(x_{il} - \overline{x_l})}{\sqrt{\sum\limits_{i=1}^{m} (x_{ik} - \overline{x_k})^2 \sum\limits_{i=1}^{m} (x_{il} - \overline{x_l})^2}} \tag{9}$$

Calculate the eigenvalues and eigenvectors of the correlation matrix $R$. The eigenvalues are denoted as $\lambda_1, \lambda_2,..., \lambda_n$ while satisfying $\lambda_i \geq 0$ (i = 1, 2,..., n), and the unitized eigenvectors corresponding to the eigenvalues are denoted as $p_1, p_2,..., p_n$.

Determine the number of principal components. The first $k$ principal components corresponding to eigenvalues greater than 1 with a cumulative contribution rate of 85%−95% are generally taken to calculate the cumulative contribution rate of the principal components.

$$v_s = \lambda_s \bigg/ \sum_{s=1}^{n} \lambda_s \quad (s = 1, 2, \cdots, n)$$

(10)

where $v_s$ is the variance contribution rate of the $s$-th principal component.

$$v_{sumk} = \sum_{s=1}^{k} \lambda_s \bigg/ \sum_{s=1}^{n} \lambda_s \quad (k = 1, 2, \cdots, n)$$

(11)

where $v_{sumk}$ is the cumulative contribution rate of the first $k$ principal components.

Calculate the corresponding scores of the extracted principal components. The principal component coefficient matrix is $U=(p_1, p_2, \ldots, p_n)$. If the first $k$ principal components are extracted from the original indicators, then:

$$y_s = X^* p_s = (x_1^*, x_2^*, \cdots, x_n^*) p_s \, (s = 1, 2, \cdots, k)$$

(12)

where $X^*$ is the standardized matrix of the original data, and $x_1^*, x_2^*, \ldots, x_n^*$, are the standardized indicator variables.

### Multicollinearity analysis

Currently, there are no universal standards for slotting performance evaluation indicators of slotted cartridge. Based on the results of "Experimental results", which studies different slotting cartridge parameters, 12 indicators for 28 samples were attained, as listed in Table 2. The data in Table 2 is converted into a correlation matrix (Table 3). When the absolute value of the correlation coefficient is greater than or equal to 0.8, it can be considered that there is a strong correlation between the indicators. When the absolute value of the correlation coefficient is between 0.6 and 0.8, it indicates that there is a strong correlation between the indicators. According to the table, taking indexes X1, X3 and X6 as examples, the correlation between index X1 and indexes X2, X4, X8 and X9 is strong. The correlation between index X3 and index X4, X8 is strong. The correlation between index X6 and index X7, X12 is strong. Therefore, the index with strong correlation can be used as a principal component by principal component analysis.

### Results

PCA was performed on the 12 indicators for the 28 samples in Table 2, and the total variance explained was displayed in Table 4. On this basis of this, a scree plot was derived, as depicted in Fig 3. The eigenvalues greater than 1 were considered as principal components. It was observed that the first three principal components account for 81.74% of the cumulative variance and are able to represent most of the information regarding the slotting performance evaluation of slotted cartridge. Therefore, the first three principal components were extracted as comprehensive evaluation indicators.

The principal component loadings (Table 4) represent the correlation coefficients between the principal components and the original variables. It can be seen from Table 5 that the first principal component F1 was significantly correlated with indicators X1, X2, X3, X4, X8, X9, X10, and X11. Therefore, F1 can be considered as a comprehensive indicator of primary crack length, secondary crack length, radius of crushed zone, primary cracks/secondary cracks, area of crushed zone, damage variable, and damage variables in slotting/non-slotting direction. The second principal component F2 was significantly correlated with indicators X6, X7, and X12, so it can be considered as a comprehensive indicator of expected slotting angle deviation, number of primary cracks, and average fractal dimensions in slotting/non-slotting direction. The third principal component F3 was correlated with indicator X5, thus it can be considered as an indicator of the number of cracks. Table 4 revealed that F1, F2, and F3 contained most of the information from 10 indicators (78.15%). Table 6 shows the load matrix of the three initial factors, and their principal component coefficients are displayed in Table 5.

**Table 2. Sample.**

| Number | X1 | X2 | X3 | X4 | X5 | X6 | X7 | X8 | X9 | X10 | X11 | X12 |
|---|---|---|---|---|---|---|---|---|---|---|---|---|
| 1 | 49.20 | 17.52 | 6.23 | 2.81 | 11 | 19 | 2 | 75.32 | 0.0264 | 1.8908 | 1.3282 | 1.0678 |
| 2 | 61.35 | 15.26 | 6.53 | 4.02 | 5 | 9 | 2 | 63.42 | 0.0291 | 2.1429 | 1.3584 | 1.0025 |
| 3 | 45.68 | 13.11 | 7.36 | 3.48 | 12 | 1 | 2 | 55.15 | 0.0267 | 1.8892 | 1.3448 | 0.9036 |
| 4 | 39.19 | 13.44 | 7.21 | 2.92 | 12 | 15 | 2 | 49.22 | 0.0224 | 1.4379 | 1.2966 | 0.9871 |
| 5 | 33.36 | 11.21 | 7.24 | 2.98 | 14 | 39 | 2 | 37.94 | 0.0181 | 1.2281 | 1.2317 | 1.0285 |
| 6 | 26.59 | 8.90 | 7.69 | 2.99 | 14 | 20 | 2 | 31.84 | 0.0155 | 1.1284 | 1.267 | 1.0191 |
| 7 | 21.02 | 7.35 | 7.9 | 2.86 | 13 | 13 | 2 | 19.35 | 0.0137 | 0.9846 | 1.1956 | 0.9827 |
| 8 | 47.21 | 21.32 | 6.52 | 2.21 | 9 | 4 | 2 | 55.01 | 0.0196 | 2.2311 | 1.3035 | 1.1146 |
| 9 | 57.01 | 17.27 | 6.42 | 3.30 | 8 | 2 | 2 | 50.95 | 0.0247 | 2.0803 | 1.3646 | 1.0465 |
| 10 | 63.20 | 20.03 | 6.66 | 3.16 | 8 | 12 | 2 | 60.81 | 0.0275 | 1.7619 | 1.4044 | 0.9545 |
| 11 | 59.35 | 24.73 | 6.11 | 2.40 | 6 | 8 | 2 | 38.74 | 0.0193 | 1.8960 | 1.3738 | 1.0171 |
| 12 | 64.81 | 21.34 | 6.68 | 3.04 | 6 | 21 | 2 | 61.65 | 0.0274 | 1.6830 | 1.4599 | 0.9449 |
| 13 | 73.85 | 21.45 | 8.58 | 3.44 | 8 | 4 | 2 | 152.73 | 0.0322 | 1.5170 | 1.5213 | 1.0363 |
| 14 | 60.28 | 19.31 | 6.71 | 3.12 | 8 | 13 | 2 | 62.91 | 0.0332 | 1.9176 | 1.3465 | 1.3732 |
| 15 | 55.13 | 17.06 | 6.42 | 3.23 | 10 | 34.5 | 4 | 50.95 | 0.0329 | 2.3291 | 1.3694 | 1.3555 |
| 16 | 62.65 | 21.32 | 5.98 | 2.94 | 6 | 49.5 | 4 | 33.8 | 0.0300 | 2.1412 | 1.3172 | 1.3585 |
| 17 | 58.85 | 18.94 | 6.37 | 3.11 | 10 | 62 | 4 | 50.95 | 0.0381 | 2.0058 | 1.3895 | 1.3592 |
| 18 | 61.49 | 20.31 | 6.53 | 3.03 | 9 | 76 | 4 | 55.42 | 0.0344 | 2.1126 | 1.3351 | 1.3786 |
| 19 | 87.46 | 17.65 | 8.14 | 4.95 | 5 | 17 | 2 | 129.62 | 0.0295 | 1.6533 | 1.2916 | 1.1352 |
| 20 | 102.56 | 19.46 | 9.45 | 5.27 | 3 | 17 | 2 | 202.01 | 0.036 | 1.6172 | 1.3636 | 1.1137 |
| 21 | 122.59 | 22.65 | 10.79 | 5.41 | 4 | 5 | 2 | 317.22 | 0.0438 | 1.6100 | 1.4216 | 1.1035 |
| 22 | 63.75 | 20.93 | 7.11 | 3.05 | 9 | 3 | 2 | 90.27 | 0.0341 | 1.8189 | 1.3590 | 0.9050 |
| 23 | 65.85 | 20.81 | 6.82 | 3.16 | 7 | 4 | 2 | 67.58 | 0.0323 | 2.0443 | 1.3133 | 0.9286 |
| 24 | 65.03 | 20.34 | 6.24 | 3.20 | 8 | 2 | 2 | 43.78 | 0.0318 | 1.9834 | 1.2910 | 0.9580 |
| 25 | 73.62 | 22.52 | 6.42 | 3.27 | 8 | 3 | 2 | 50.95 | 0.0379 | 2.1475 | 1.3424 | 0.9778 |
| 26 | 80.85 | 19.56 | 6.76 | 4.13 | 10 | 7 | 2 | 65.02 | 0.0349 | 2.2158 | 1.3321 | 1.0246 |
| 27 | 89.73 | 21.33 | 6.62 | 4.21 | 8 | 4 | 2 | 59.14 | 0.0308 | 2.6019 | 1.3472 | 1.0375 |
| 28 | 97.02 | 23.02 | 6.93 | 4.21 | 8 | 1 | 2 | 72.33 | 0.0356 | 1.1704 | 1.2891 | 1.0583 |

**Table 3. Correlation matrix.**

| Index | X1 | X2 | X3 | X4 | X5 | X6 | X7 | X8 | X9 | X10 | X11 | X12 |
|---|---|---|---|---|---|---|---|---|---|---|---|---|
| X1 | 1 | 0.70 | 0.45 | 0.81 | −0.78 | −0.22 | −0.08 | 0.75 | 0.80 | 0.22 | 0.45 | 0.08 |
| X2 | 0.70 | 1 | −0.11 | 0.15 | −0.70 | −0.12 | 0.09 | 0.32 | 0.63 | 0.50 | 0.57 | 0.14 |
| X3 | 0.45 | −0.11 | 1 | 0.68 | −0.24 | −0.20 | −0.30 | 0.86 | 0.22 | −0.49 | 0.17 | −0.12 |
| X4 | 0.81 | 0.15 | 0.68 | 1 | −0.54 | −0.20 | −0.18 | 0.75 | 0.57 | −0.05 | 0.17 | −0.01 |
| X5 | −0.79 | −0.70 | −0.24 | −0.54 | 1 | 0.16 | 0.03 | −0.57 | −0.56 | −0.32 | −0.50 | −0.11 |
| X6 | −0.22 | −0.12 | −0.20 | −0.20 | 0.16 | 1 | 0.85 | −0.20 | 0.06 | 0.06 | −0.07 | 0.72 |
| X7 | −0.08 | 0.09 | −0.30 | −0.18 | 0.03 | 0.85 | 1 | −0.19 | 0.27 | 0.34 | 0.07 | 0.80 |
| X8 | 0.75 | 0.32 | 0.86 | 0.75 | −0.57 | −0.20 | −0.19 | 1 | 0.56 | −0.14 | 0.45 | 0.02 |
| X9 | 0.80 | 0.63 | 0.22 | 0.57 | −0.56 | 0.06 | 0.27 | 0.56 | 1 | 0.38 | 0.49 | 0.30 |
| X10 | 0.22 | 0.50 | −0.49 | −0.05 | −0.32 | 0.06 | 0.34 | −0.14 | 0.38 | 1 | 0.29 | 0.28 |
| X11 | 0.45 | 0.57 | 0.18 | 0.17 | −0.50 | −0.07 | 0.07 | 0.45 | 0.49 | 0.29 | 1 | 0.05 |
| X12 | 0.08 | 0.14 | −0.12 | −0.01 | −0.11 | 0.72 | 0.80 | 0.02 | 0.3 | 0.28 | 0.05 | 1 |

**Table 4. Total variance explained for PCA.**

| Principal component number | 1 | 2 | 3 | 4 | 5 | 6 | 7 | 8 | 9 | 10 | 11 | 12 |
|---|---|---|---|---|---|---|---|---|---|---|---|---|
| Eigenvalue | 4.796 | 3.090 | 1.830 | 0.754 | 0.465 | 0.380 | 0.266 | 0.180 | 0.119 | 0.093 | 0.025 | 0.002 |
| Cumulative variance/% | 39.96 | 65.71 | 80.97 | 87.25 | 91.13 | 94.29 | 96.51 | 98.01 | 98.99 | 99.78 | 99.98 | 100 |

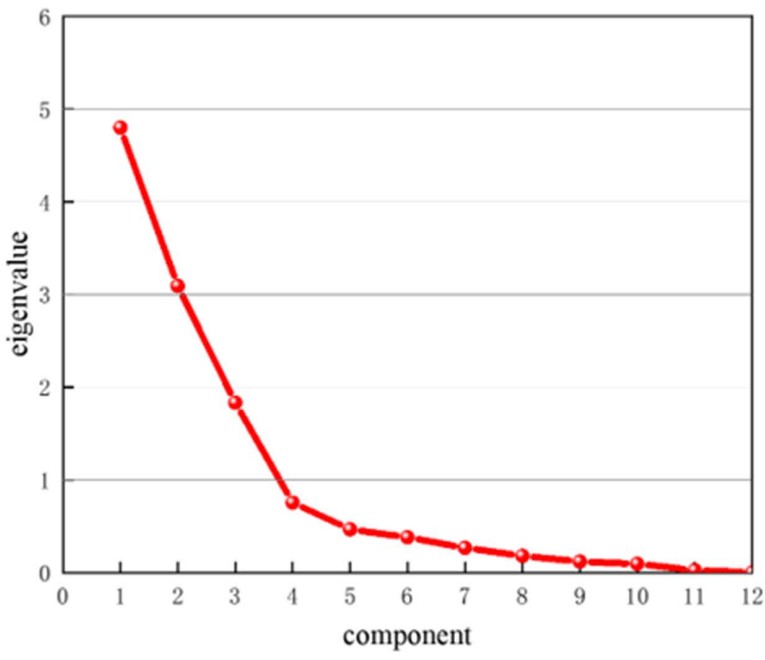

**Fig 3. Scree plot for PCA.**

**Table 5. Component matrix.**

| Index | F1 | F2 | F3 |
|---|---|---|---|
| X1 | 0.960 | 0.005 | −0.019 |
| X2 | 0.680 | 0.365 | −0.490 |
| X3 | 0.536 | −0.544 | 0.583 |
| X4 | 0.775 | −0.274 | 0.340 |
| X5 | −0.823 | −0.122 | 0.196 |
| X6 | −0.219 | 0.730 | 0.549 |
| X7 | −0.074 | 0.888 | 0.362 |
| X8 | 0.842 | −0.280 | 0.346 |
| X9 | 0.808 | 0.346 | 0.028 |
| X10 | 0.228 | 0.622 | −0.529 |
| X11 | 0.610 | 0.199 | −0.245 |
| X12 | 0.088 | 0.791 | 0.450 |

**Table 6. Coefficient matrix of the three principal components.**

| Index | F1 | F2 | F3 |
|---|---|---|---|
| X1 | 0.739 | 0.612 | −0.050 |
| X2 | 0.914 | −0.020 | −0.012 |
| X3 | −0.113 | 0.945 | −0.136 |
| X4 | 0.302 | 0.834 | −0.065 |
| X5 | −0.773 | −0.361 | 0.049 |
| X6 | −0.155 | −0.091 | 0.922 |
| X7 | 0.114 | −0.182 | 0.937 |
| X8 | 0.347 | 0.884 | −0.069 |
| X9 | 0.736 | 0.402 | 0.263 |
| X10 | 0.695 | −0.444 | 0.196 |
| X11 | 0.667 | 0.159 | −0.005 |
| X12 | 0.152 | 0.020 | 0.901 |

The expressions for the principal components F1, F2, and F3 can be derived from Table 6 as:

$$F1 = 0.739X1 + 0.914X2 - 0.113X3 + 0.302X4 - 0.773X5 - 0.155X6 + 0.114X7 + 0.347X8 + 0.736X9 + 0.695X10 + 0.667X11 + 0.152X12 \tag{13}$$

$$F2 = 0.612X1 - 0.020X2 + 0.945X3 + 0.834X4 - 0.361X5 - 0.091X6 - 0.182X7 + 0.884X8 + 0.402X9 - 0.444X10 + 0.159X11 + 0.020X12 \tag{14}$$

$$F3 = -0.050X1 - 0.012X2 - 0.136X3 - 0.065X4 + 0.049X5 + 0.922X6 + 0.937X7 - 0.069X8 + 0.263X9 + 0.196X10 - 0.005X11 + 0.901X12 \tag{15}$$

Substituting the data from Table 2 into Eqs. (13)-(15) to calculate the scores of principal components, and the results are presented in Table 7.

The scatter points of the principal components F1-F3 are illustrated in Figs 4−6, with sample numbers corresponding to the order of the slotted cartridge experiments. The data in Table 7 aligns with the information in Figs 4−6. The plots of PCA scores reflect the relationship between the objects and the indicators. Fig 4 revealed that F1 and F2 account for 39.96% and 25.75% of the original data, respectively. Samples 2, 13, 19, 22, 26, 27, and 28 fell in the first quadrant. Fig 4 and Tables 4–6 demonstrated that the first principal component serves as a comprehensive indicator for primary crack length, secondary crack length, radius of crushed zone, primary cracks/secondary cracks, area of crushed zone, damage variable, and average damage variables in slotting/non-slotting direction. Conversely, samples 1, 3, 4, 5, 6, 7 and 8 were distributed in the negative direction of F1 and F2, i.e., they were in the third quadrant. The samples 1, 3, 4, 5, 6, and 7 exhibited lower values in F1. This suggested that when the decoupling coefficients were 1.50 and 1.83–3.33, F1 decreased with increasing decoupling coefficient. Additionally, for decoupling coefficients of 1.50 and 1.83–2.53. Given that these 12 samples performed poorly in the first principal component, and due to the significant contribution of the variance of the first principal component, they were mostly dominated by the poorer outcomes for crack length and damage. Therefore, when the decoupling coefficient was between 1.73 and 2.50, the outcomes for crack length and damage were poorer, while slotting angles between 90° and 165° yielded better results. This led to the conclusion that the importance of slotting angle on the slotting performance is greater than that of decoupling coefficient. Samples 15, 16, 17, and 18 were positioned in the fourth quadrant, where

**Table 7. PCA scores of each component.**

| Numbering | F1 | F2 | F3 |
|---|---|---|---|
| 1 | −1.8976 | −2.3094 | −0.1161 |
| 2 | 1.0680 | 0.1444 | −1.0021 |
| 3 | −3.0276 | −0.9792 | −2.1100 |
| 4 | −5.1651 | −1.8673 | −1.2601 |
| 5 | −8.0492 | −2.6119 | −0.0460 |
| 6 | −8.7778 | −2.3232 | −1.1890 |
| 7 | −10.3129 | −2.5575 | −1.8890 |
| 8 | −1.1515 | −3.5478 | −0.5142 |
| 9 | −0.1923 | −1.5240 | −1.0917 |
| 10 | 0.5276 | −0.6069 | −1.3590 |
| 11 | 0.7789 | −2.7427 | −1.1837 |
| 12 | 1.6976 | −0.2360 | −1.0783 |
| 13 | 2.9989 | 3.8752 | −1.6960 |
| 14 | 0.9494 | −0.6027 | 1.4876 |
| 15 | 0.8228 | −2.3445 | 5.4083 |
| 16 | 1.6251 | −2.7389 | 5.9508 |
| 17 | 1.2437 | −1.8488 | 6.6814 |
| 18 | 1.0375 | −2.0576 | 7.4048 |
| 19 | 1.5594 | 4.6071 | −0.5481 |
| 20 | 4.6973 | 8.3853 | −0.8806 |
| 21 | 7.7949 | 12.4324 | −1.6688 |
| 22 | 0.8709 | 0.2108 | −1.9406 |
| 23 | 1.1694 | −0.4148 | −1.6284 |
| 24 | 0.3686 | −1.3765 | −1.4822 |
| 25 | 2.6143 | −0.5406 | −1.1051 |
| 26 | 1.7781 | 0.5489 | −0.7665 |
| 27 | 3.4694 | 0.3010 | −0.7266 |
| 28 | 1.5022 | 2.7251 | −1.6507 |

F1 was positive but small, and F2 was negative and the smallest among all samples. The principal component F was a comprehensive indicator of expected slotting angle deviation, number of primary cracks, and average fractal dimensions in slotting/non-slotting direction, indicating that F2 was the worst when the slotting width was greater than 0.5 mm. Due to the dense points near the origin (0,0), this area was magnified for easier observation and analysis, as depicted in Fig 4(b). Under principal component F1, samples 11, 12, and 13 represent different slotting tube materials, while samples 8, 9, and 10 represent different slotting tube thicknesses. The average values of different decoupling coefficients under comprehensive indicator F1 were calculated as −5.17 for decoupling coefficient, −0.27 for slotting tube thickness, 1.83 for slotting tube material, 1.14 for slotting width, 4.68 for charge amount, and 1.68 for slotting angle. Thus, the order of importance in comprehensive indicator F1 was charge amount > tube material > slotting angle > slotting width > tube thickness > decoupling coefficient. Similarly, the average values of different decoupling coefficients under comprehensive indicator F2 were −1.78, −1.89 for tube thickness, 0.30 for tube material, −1.92 for slotting width, 8.47 for charge amount, and 0.21 for slotting angle. Therefore, the order of importance of te parameters affecting the slotting performance in comprehensive indicator F2 was charge amount > slotting tube material > slotting angle > decoupling coefficient > slotting tube thickness > slotting width.

A

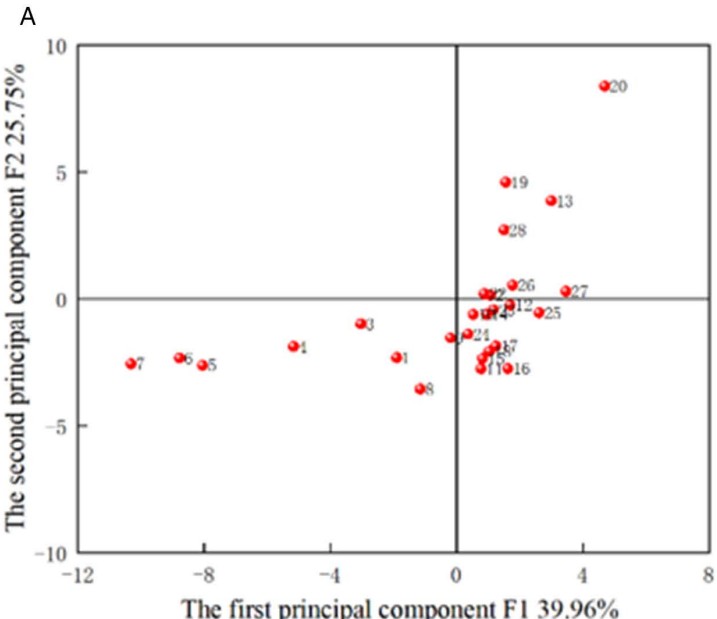

B

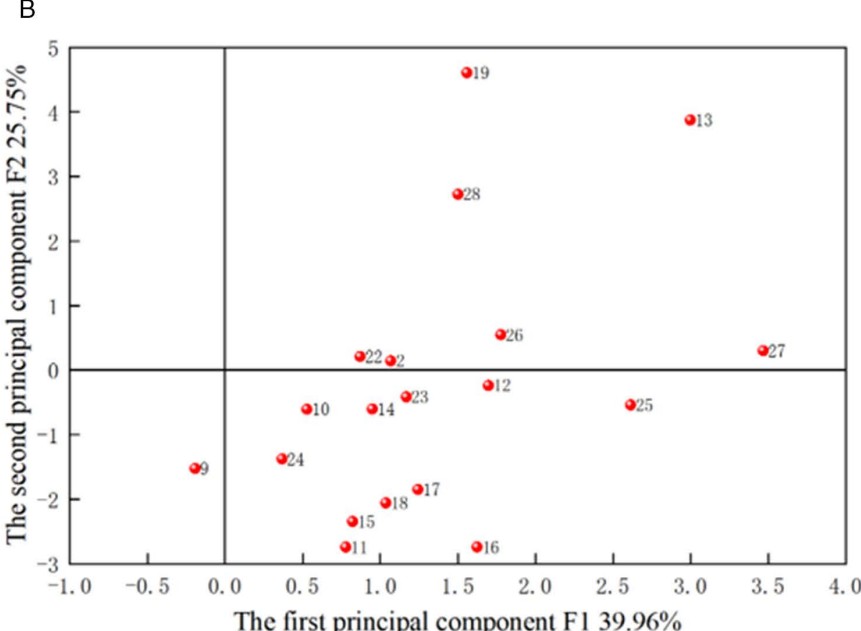

**Fig 4. PCA scores of F1 and F2.** (a) Overall, (b) Localized magnified view.

Fig 5 depicted that F1 and F3 account for 39.96% and 15.25% of the original data, respectively, totaling 55.21%. Samples 15, 16, 17 and 18 fell in the positive direction of F1 and F3, i.e., in the first quadrant, indicating that the slotting width performed well in both F1 and F3, where F3 is the comprehensive indicator of the number of cracks. This suggested that samples 15−18 exhibited favorable results in terms of primary crack length, secondary crack length, radius of crushed zone, primary cracks/secondary cracks, area of crushed zone, damage variable, and average damage variables in slotting/non-slotting direction, with fewer cracks. Samples 1, 3−8 were positioned in the negative direction of F1 and

A

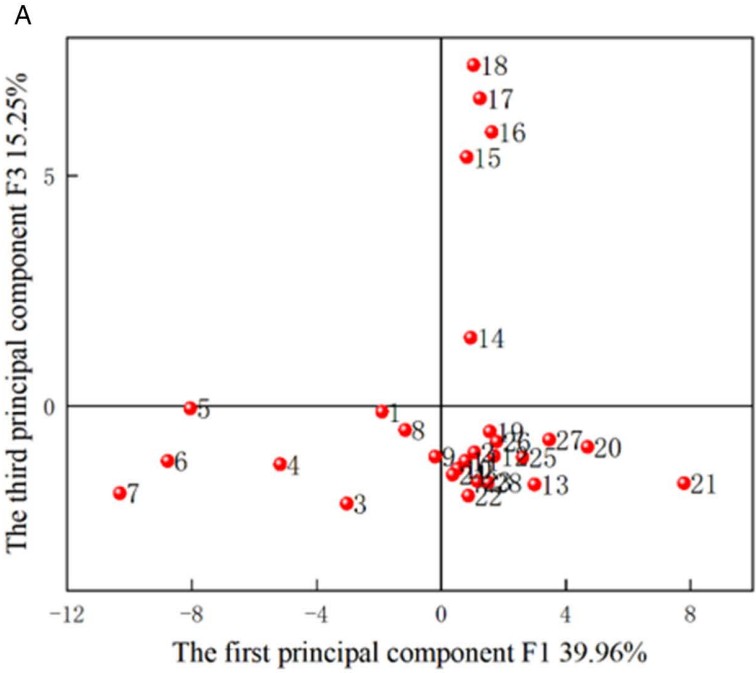

B

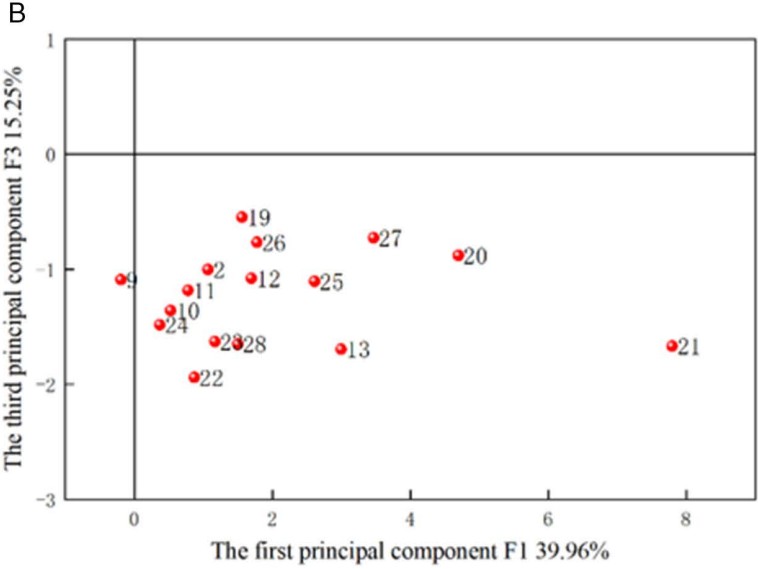

**Fig 5. PCA scores of F2 and F3.** (a) Overall, (b) Localized magnified view.

F3, i.e., in the third quadrant, indicating that the decoupling coefficient adversely affected these indicators, leading to a higher number of cracks. By magnifying the remaining sample data in Fig 5(b), the average values of different decoupling coefficients in the comprehensive indicator F3 were calculated as −1.09 for decoupling coefficient, −0.99 for slotting tube thickness, −1.32 for slotting tube material, 5.39 for slotting width, −1.03 for charge amount, and −1.33 for slotting angle. Therefore, the order of importance of the parameters affecting the slotting performance in the comprehensive indicator F3 was slotting width > slotting tube thickness > charge amount > decoupling coefficient > slotting angle > slotting tube material.

Fig 6 indicated that F2 and F3 account for 25.75% and 15.25% of the original data, respectively, totaling 41%, which was below the typical information threshold. Samples 1, 3, 4, 5, 6, 7, 8, 9, 10, 11, 12, 23, 24 and 25 fell in the negative direction of F2 and F3, i.e., in the third quadrant, suggesting that decoupling coefficient, slotting width, and slotting angle performed poorly in both F2 and F3. This indicated that these 16 samples perform inadequately in terms of expected slotting angle deviation, number of primary cracks, average fractal dimensions in slotting/non-slotting direction, and number of cracks.

A

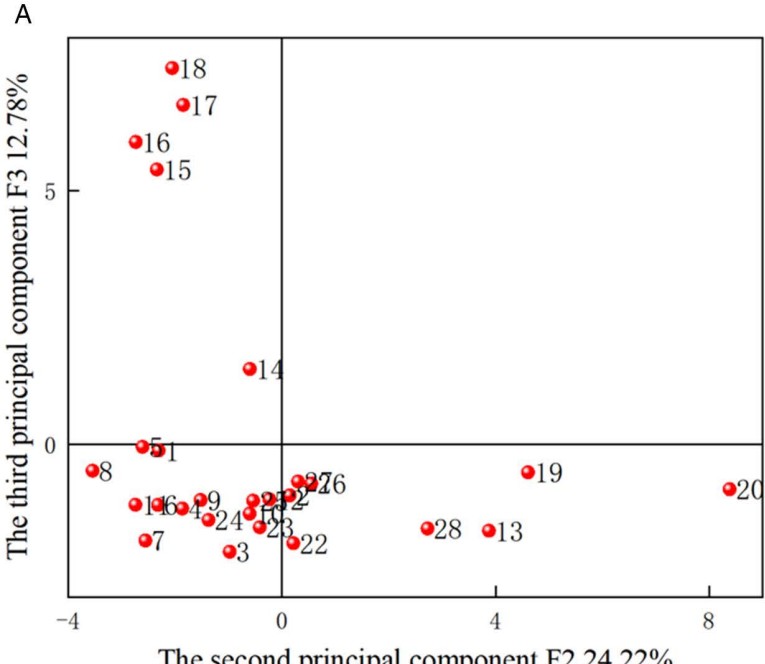

B

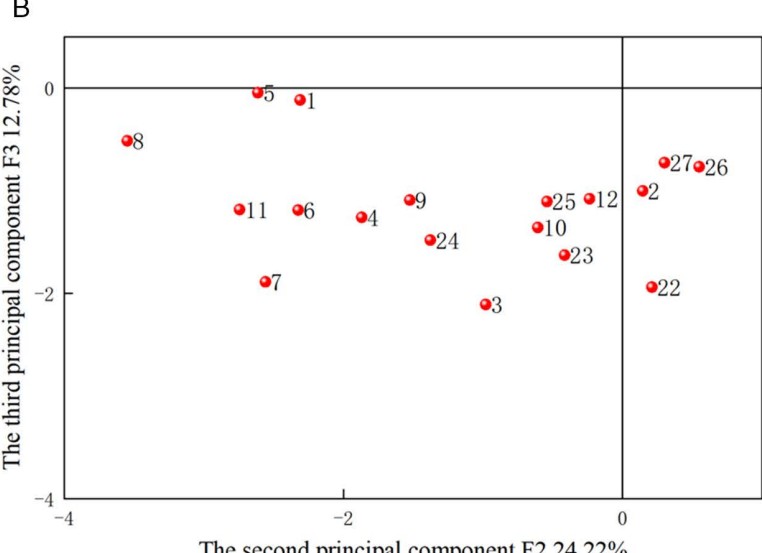

**Fig 6. PCA scores of F1 and F3.** (a) Overall, (b) Localized magnified view.

These analyses indicated that the scatter plots of F1 versus F2, F1 versus F3, and F2 versus F3 can provide valuable insights for qualitatively assessing the slotting performance of slotted cartridge.

A composite evaluation function (Eq. 16) was employed to quantitatively analyze the slotting performance levels of 28 slotted cartridge samples.

$$F = \eta_1 F1 + \eta_2 F2 + \eta_3 F3 \tag{16}$$

where $\eta_1$, $\eta_2$, and $\eta_3$ are the variance contribution rates of F1, F2, and F3. Substituting Eqs. (13)-(15) into Eq. (16) yields:

$$F = 0.3996F1 + 0.2575F2 + 0.1525F3 \tag{17}$$

The comprehensive principal component results can be calculated by Eq. (17) (see Table 8). It was observed that samples with good slotting performance have comprehensive principal component values ranging from [0, 6.0617], which includes samples 2, 12, 13, 14, 15, 16, 17, 18, 19, 20, 21, 22, 23, 25, 26, 27 and 28. Fig 2 indicated that the evaluation results for these samples are consistent with their actual performances. Samples exhibiting poor slotting performance have comprehensive principal component values ranging from [−5.0677, 0], which encompasses samples 1, 3, 4, 5, 6, 7, 8, 9, 10, 11 and 24, and their evaluation results align with actual performance. Given that the PCA score range is [−5.0677, 6.0617], the comprehensive principal component results can be categorized into four slotting performance levels. The comprehensive principal component score is divided into two kinds of cutting effects (G1 and G2) on average in the interval of poor cutting effect [−5.0677, 0]. The G1 level indicates poor slotting performance, with comprehensive principal component values ranging from [−5.0677, −2.5339], including samples 4, 5, 6, and 7. The G2 level shows below-average slotting performance, with values ranging from [−2.2853, 0], encompassing samples 1, 3, 8, 9, 10, 11 and 24. The G3 level exhibits average performance, with values from [0, 3.0309], including samples 2, 12, 13, 14, 15, 16, 17, 18, 19, 22, 23, 25, 26, 27 and 28. The G4 level demonstrates excellent performance, with values ranging from [3.0309, 6.0617], which includes samples 20 and 21.

## Order of importance of the different parameters of slotted cartridge

A quantitative analysis of the slotting performance levels of the decoupling coefficient across different comprehensive indicators F1, F2, and F3 was conducted using the composite evaluation function (Eq. 19). Likewise, the scores of slotting tube thickness, slotting tube material, slotting width, charge amount, and slotting angle were determined, and the results are listed in Table 9.

**Table 8. Comprehensive principal components.**

| Scheduling | 1 | 2 | 3 | 4 | 5 | 6 | 7 |
|---|---|---|---|---|---|---|---|
| F | 6.0617 | 3.9020 | 1.9376 | 1.7259 | 1.3531 | 1.0503 | 1.0398 |
| Numbering | 21 | 20 | 13 | 19 | 27 | 17 | 18 |
| Scheduling | 8 | 9 | 10 | 11 | 12 | 13 | 14 |
| F | 1.0140 | 0.8516 | 0.7370 | 0.7350 | 0.5498 | 0.4532 | 0.4510 |
| Numbering | 28 | 16 | 25 | 26 | 15 | 12 | 14 |
| Scheduling | 15 | 16 | 17 | 18 | 19 | 20 | 21 |
| F | 0.3111 | 0.1122 | 0.1063 | −0.1527 | −0.4332 | −0.5755 | −0.6357 |
| Numbering | 2 | 23 | 22 | 10 | 24 | 11 | 9 |
| Scheduling | 22 | 23 | 24 | 25 | 26 | 27 | 28 |
| F | −1.3706 | −1.4521 | −1.7838 | −2.7370 | −3.8961 | −4.2872 | −5.0677 |
| Numbering | 1 | 8 | 3 | 4 | 5 | 6 | 7 |

**Table 9. Scores of different parameters.**

| Sample set | Decoupling coefficient | Slotting tube thickness | Slotting tube material | Slotting width | Charge amount | Slotting angle |
|---|---|---|---|---|---|---|
| F1 | −5.166 | −0.272 | 1.825 | 1.136 | 4.684 | 1.682 |
| F2 | −1.786 | −1.893 | 0.299 | −1.919 | 8.475 | 0.208 |
| F3 | −1.087 | −0.988 | −1.319 | 5.387 | −1.033 | −1.329 |
| F | −2.690 | −0.747 | 0.605 | 0.781 | 3.897 | 0.523 |

The comprehensive principal component results from the various sample sets indicated that the order of importance of the parameters affecting the slotting performance was charge amount > slotting width > tube material > slotting angle>slotting tube thickness > decoupling coefficient. For individual samples, the slotting performance is good for F > 0 and poor for F < 0. Therefore, it is recommended to prioritize the order of decoupling coefficient, slotting tube thickness, slotting angle, slotting tube material, slotting width, and charge amount in determining the parameters for slotted cartridge.

## PCA-PNN model

### PNN model

Probabilistic Neural Network (PNN) is a neural network based on the radial basis functions and the classic principle of probability density estimation. Its network structure is illustrated in Fig 7 [43]. The algorithm steps of PNN are:

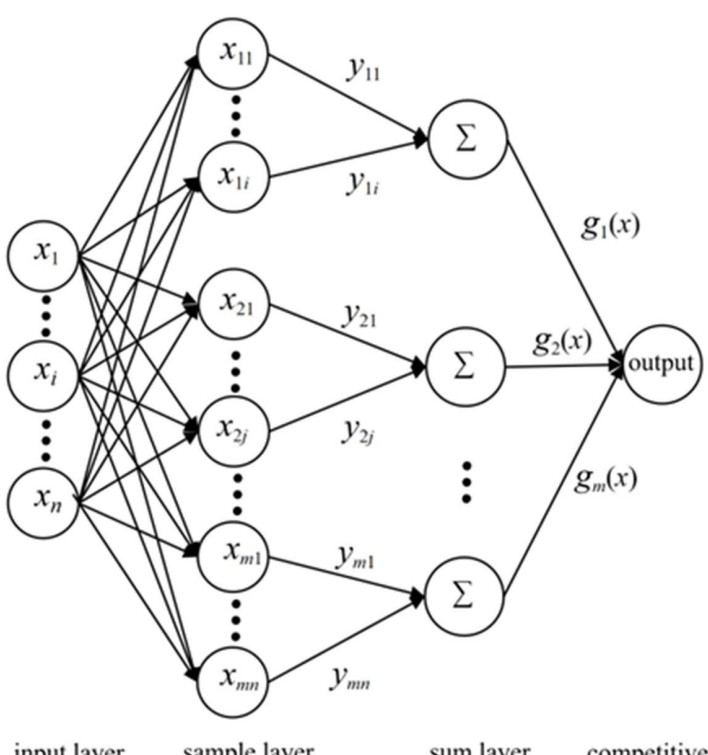

**Fig 7. PNN topology.**

First, input the sample vector $X$ to be tested into the input layer, where the number of neurons equals the sample dimensions. The sample layer calculates the distance between the sample vector $X$ to be tested and the training samples. The output of each node unit in this layer is calculated as:

$$f(X, W_i) = exp[-(X - W_i)^T(X - W_i)/2\delta^2]$$

(18)

where $X$ is the sample vector to be tested; $W_i$ is the weight from the input layer to the sample layer; $\delta$ is the smoothing parameter.

Then, the summation layer performs the summation of the probability density function (PDF) of a certain class. The PDF estimates of each class can be obtained by the Parzen window method:

$$f_A(X) = \frac{1}{((2\pi)^{(p/2)}\delta^p)} \frac{1}{m} \sum_{j=1}^{m} exp((X - X_{ai})^T(X - X_{ai})/2\delta^2)$$

(19)

where $f_A(X)$ is the probability density function value of the class; $X_{ai}$ is the $i$ training sample vectors; $m$ is the number of training samples; $p$ is the dimensions of the sample vector $X$ to be tested and the training sample vector $X_{ai}$.

Finally, the competition layer outputs the PDFs of each class. The class with the maximum probability is 1, and the other classes are 0.

## Determination of PNN network parameters

The dimensionality reduction results via PCA yielded three principal components F1, F2, and F3, which contain the majority of the information from the original indicators. Thus, these can serve as comprehensive predictive indicators RCI1, RCI2, and RCI3 for the slotting performance of slotted cartridges. The comprehensive predictive indicators RCI1, RCI2, and RCI3 for each group of slotted cartridges constitute the input vectors for the PNN. The score data for the principal components in the Table 7 were utilized as new sample data for PNN simulation, and there were 28 groups of training samples. The four slotting performance levels, G1, G2, G3, and G4, derived from the earlier analysis served as input vectors for the PNN model. The expected output values for the neurons in the competitive layer of the PNN model were set to 1(G1), 2(G2), 3(G3), and 4(G4). Here, G1, G2, G3, and G4 represent the slotting performance levels of slotted cartridge. Consequently, the numbers of neurons in the input layer, pattern layer, accumulation layer, and output layer of the PNN were 3, 28, 4, and 4, respectively, resulting in a topological network structure of $3 \times 28 \times 4 \times 4$.

The selection of the smoothing factor is critical for network performance [13]. The value of the smoothing factor Spread was taken as a uniformly distributed interval [0.02,1.00], thus resulting in 50 Spread values. After testing, the variation of the prediction accuracy for training and test samples of slotted cartridge with Spread values is illustrated in Fig 8. As depicted in Fig 8, the maximum correctness of test samples and the first maximum were reached at a Spread value of 0.1. and the correctness was highest at a Spread value of 0.1 in the training samples.

## Prediction results of the PCA-PNN model

Furthermore, to assess the generalization capability of the PNN model and ensure the reliability of the evaluation results for the slotting performance levels of slotted cartridge, it is essential to avoid a subjective selection of training samples that are abundant while having few test samples. The 28 samples listed in Table 8 were organized into nine different configurations for training, with the ratios of training to test samples set at 26:2, 24:4, 22:6, 20:8, 18:10, 16:12, 14:14, and 12:16. This setup aims to investigate the impact of the number of training samples on the predictive accuracy of the PNN model. A Spread value of 0.1 was set to ensure accurate evaluations of the slotting performance levels of slotted cartridge. Simultaneously, the sample data (see Table 2) without PCA were also organized into the aforementioned eight configurations for training.

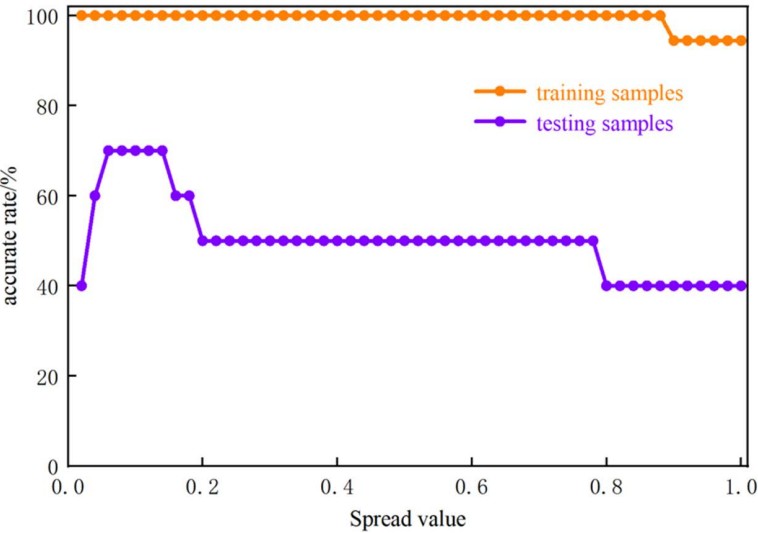

**Fig 8. Variation of model prediction correctness with Spread values.**

Due to space constraints, only the scenario with a ratio of training to test samples of 20:8 is included (see Fig 9). Fig 9(a) illustrated that after dimensionality reduction via PCA, the PCA-PNN model trained on samples 1–20 predicted them correctly without errors. However, for test samples 21–28, samples 21, 25, and 27 were incorrectly predicted, yielding a correct prediction rate of 96.43% for the PNN model after PCA. Fig 9(b) demonstrated that for the sample data without PCA (i.e., raw data), the PNN model results indicated no errors in predicting the training samples, but five test samples

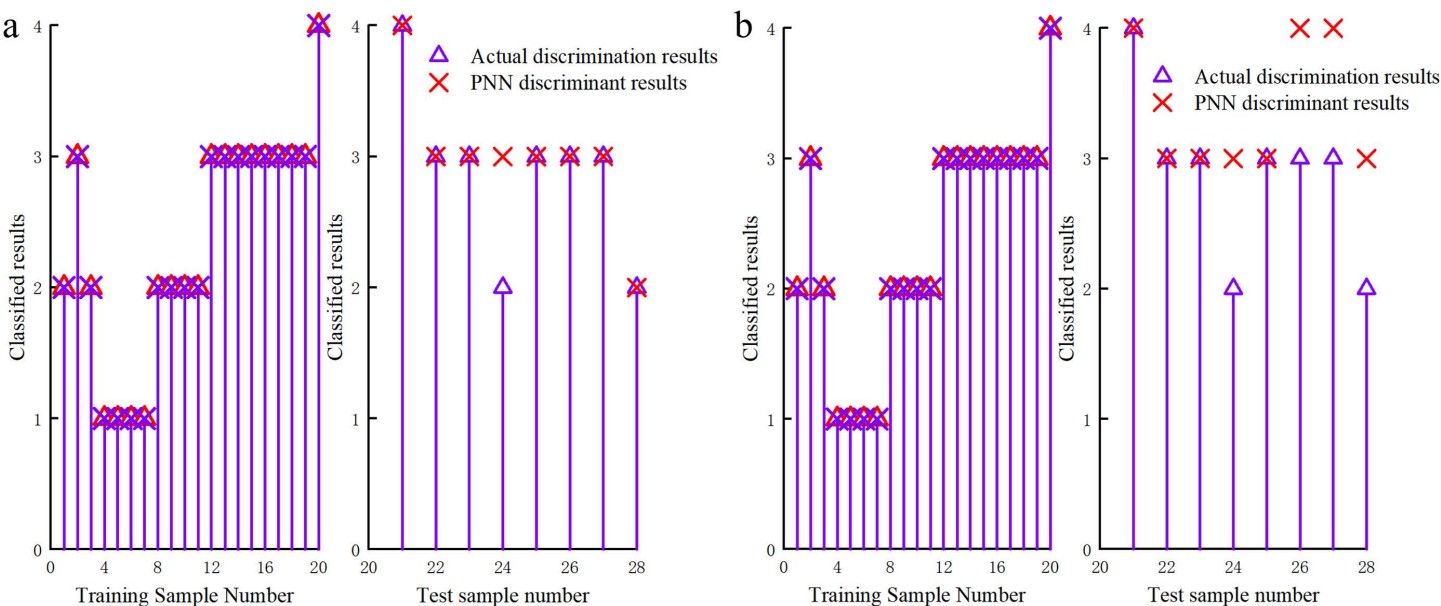

**Fig 9. (a) Test effect of slotted cartridge samples with PCA; (b) Test effect of slotted cartridge samples without PCA.**

were incorrectly predicted, resulting in a correct prediction rate of 85.71% for the PNN model without PCA. The predictive accuracy of the PNN model after PCA increased by 10.72%.

The statistical results of the PNN model prediction for ratios of training to test samples of 26:2, 24:4, 22:6, 20:8, 18:10, 16:12, 14:14, 12:16, and 10:18 are presented in Fig 10 and Tables 10, 11. Fig 10 indicated that as the number of training

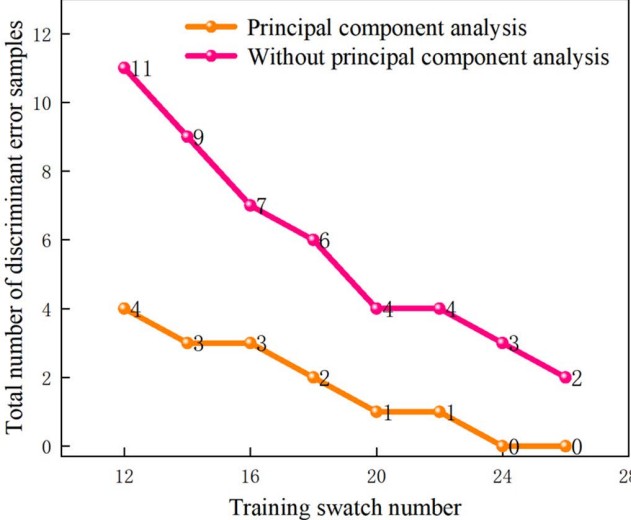

**Fig 10. Effect of the number of training samples on the PNN model.**

**Table 10. Results of eight learning configurations for PNN model with PCA.**

| Ratios of training to test samples | Number of training error | Number of test error | Correct prediction rate of PCA-PNN/% |
|---|---|---|---|
| 26:2 | None | | 100 |
| 24:4 | None | | 100 |
| 22:6 | None | 24 | 96.43 |
| 20:8 | None | 24 | 96.43 |
| 18:10 | None | 20, 24 | 92.86 |
| 16:12 | None | 17, 20, 24 | 89.29 |
| 14:14 | None | 17, 20, 24 | 89.29 |
| 12:16 | None | 12, 17, 20, 24 | 85.71 |

**Table 11. Results of eight learning configurations for PNN model without PCA.**

| Ratios of training to test samples | Number of training error | Number of test error | Correct prediction rate of PNN/% |
|---|---|---|---|
| 26:2 | None | 27, 28 | 92.86 |
| 24:4 | None | 26, 27, 28 | 89.29 |
| 22:6 | None | 24, 26, 27, 28 | 85.71 |
| 20:8 | None | 24, 26, 27, 28 | 85.71 |
| 18:10 | None | 19, 24, 26, 27, 28 | 82.14 |
| 16:12 | None | 18, 19, 24, 26, 27, 28 | 78.57 |
| 14:14 | None | 15, 16, 17, 18, 19, 24, 26, 27, 28 | 67.86 |
| 12:16 | None | 13, 15, 16, 17, 18, 19, 24, 26, 27, 28 | 64.29 |

samples decreased, the predictive accuracy of the PNN model declined. Fig 10 and Tables 10, 11 revealed that the PCA-PNN model exhibited robust predictive performance across the eight training and test sample configurations, achieving correct prediction rates of 100%, 100%, 96.43%, 96.43%, 92.86%, 89.29%, 89.29%, 85.71%. The average accuracy was improved by 12.95% compared to that of the data without PCA dimensionality reduction. The evaluation results of the PNN model after PCA were in agreement with the actual situations, as illustrated in Table 2, which demonstrates the feasibility of the PNN model in assessing the slotting performance levels of slotted cartridge after PCA.

## Conclusions

(1) When the decoupling coefficient is 1.67, the slitting effect is the best. The main crack length is positively correlated with the slotting tube thickness, but the crack length increases slowly with the increase of the tube thickness. The slotting tube material has a great influence on the slitting effect. The effect of iron tube is the best, the effect of aluminum pipe is in the middle, and the effect of PVC pipe is the worst. For the slotting width, when the slotting width ≥ 0.75 mm, two main cracks with angle will be produced in the slotting direction. As the slotting width increases, the angle between the main cracks increases. The larger the charge amount, the longer the main crack; Different cutting angles, the crack can basically achieve the purpose of directional fracture; the longest crack decreases with the increase of the slotting angle, and the above conclusions are consistent with the related slotting cartridge research.

(2) Given the numerous factors influencing the slotting performance of slotted cartridge, the score $F$ of comprehensive principal component was derived through PCA. The consistency of the predicted slotting performances with those acquired from the dynamic caustics experiment was judged by the comprehensive principal components $F > 0$ and $F < 0$, which proves the reliability of the PCA. The F values of comprehensive principal components for various parameters were computed and ranked, which revealed that the order of importance of the factors affecting the slotting performance of slotted cartridge was decoupling coefficient, slotting tube thickness, slotting angle, slotting tube material, slotting width, and charge amount.

(3) After dimensionality reduction through PCA, the predictive accuracy of the PNN model decreased as the number of training samples decreased. The PCA-PNN model demonstrated good predictive performance across eight different training and test sample configurations, achieving correct prediction rates of 100%, 100%, 96.43%, 96.43%, 92.86%, 89.29%, 89.29% and 85.71%. The average accuracy was improved by 12.95% compared to that of the data without PCA dimensionality reduction, which demonstrates that the combination of PCA and PNN model is feasible for evaluating the slotting performance levels of slotted cartridge.

## Acknowledgments

The authors would like to acknowledge the efforts made by Cheng Fang and Yongxin Yao in their assistance in writing the manuscript.

## Author contributions

**Conceptualization:** Qiang Li.

**Data curation:** Qiang Li, Jinshan Sun, Nan Jiang.

**Formal analysis:** Qian Dong.

**Funding acquisition:** Jinshan Sun.

**Investigation:** Xianqi Xie.

**Methodology:** Jianguo Wang, Nan Jiang.

Project administration: Jianguo Wang.

Resources: Xianqi Xie, Jianguo Wang.

Visualization: Qian Dong.

Writing – original draft: Qiang Li.

Writing – review & editing: Jinshan Sun.

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
