## [Decision Letter · Decision Letter 0]

17 Jul 2025

PONE-D-25-31588Slotting blasting model experiment and PCA-PNN evaluation model of influencing factors of slitting effectPLOS ONE

Dear Dr. sun,

Thank you for submitting your manuscript to PLOS ONE. After careful consideration, we feel that it has merit but does not fully meet PLOS ONE’s publication criteria as it currently stands. Therefore, we invite you to submit a revised version of the manuscript that addresses the points raised during the review process.

The reviewers have recommended a major revision. Kindly submit a detailed point-by-point rebuttal and ensure that all changes are clearly highlighted in the revised manuscript. 

We look forward to receiving your revised manuscript.

Kind regards,

Mithilesh K. Dikshit

Academic Editor

PLOS ONE

 [The National Key Research and Development Program (2021-008), Wuhan key Research and Development Program (2024050802030155), 2024 Chutian Talent Plan—Science and Technology Innovation Team Project.]. 

Additional Editor Comments (if provided):

Reviewers' comments:

Reviewer's Responses to Questions

**Comments to the Author**

1. Is the manuscript technically sound, and do the data support the conclusions?

Reviewer #1: Yes

Reviewer #2: No

Reviewer #3: Yes

2. Has the statistical analysis been performed appropriately and rigorously? 

Reviewer #1: Yes

Reviewer #2: No

Reviewer #3: Yes

3. Have the authors made all data underlying the findings in their manuscript fully available?

Reviewer #1: Yes

Reviewer #2: Yes

Reviewer #3: Yes

4. Is the manuscript presented in an intelligible fashion and written in standard English?

Reviewer #1: Yes

Reviewer #2: No

Reviewer #3: Yes

5. Review Comments to the Author

Reviewer #1: The manuscript is well-structured and presents original research with high relevance to engineering blasting applications. The hybrid PCA-PNN approach is effective and validated rigorously. Therefore, I recommend acceptance after minor revisions, primarily for language clarity , Revising References and enhanced explanation of classification thresholds and model selection rationale.

Reviewer #2: The structure of the paper is not well enough to publish in this journal. Without a proper modification, the paper is not acceptable. Here are some strict and challenging comments for the authors to address:

1. The grammar of the paper needs major revisions.

2. The abstract should illustrate the novelty clearly; however, the structure of the abstract is not well.

3. The literature review is not good. It is recommended to use these works to enhance the literature review. These works can illustrate the application of the matter in the real world:

• Polymer electrolyte membrane fuel cell performance Revolutionized: Artificial intelligence-validated asymmetric flow channels enhance mass transport via hybrid analytical-numerical frameworks

• Seismic inversion based on principal component analysis and probabilistic neural network for prediction of porosity from post-stack seismic data

• Permanent magnet synchronous motor demagnetization fault diagnosis and localization

• Smart Design of Conical Vortex Generators for Heat Transfer Enhancement: A Synergy of CFD and Bio-Inspired Optimization

4. The paper assumes PCA is suitable for dimensionality reduction without explicitly testing for multicollinearity among the 12 indicators.

5. While the authors claim that PCA was used to reduce dimensionality and eliminate correlation among indicators, there is no statistical evidence provided that such correlations were problematic in the first place. A correlation matrix or VIF values would have supported this decision.

6. The classification of slotting performance into four levels (G1–G4) based on PCA scores appears arbitrary. There is no explanation of how the thresholds (e.g., [-5.0677, 2.5339] for G1) were determined or whether they correspond to meaningful physical differences in slitting performance.

7. The PNN model uses 28 samples with 3 input features (after PCA). While the reported accuracy is high, the small sample size raises concerns about overfitting, especially since some classes (e.g., G1 and G4) contain very few samples.

Reviewer #3: interesting work . but some corrections are needed

specify the novelty through the text

lack of related references. the following references are recommended to ad to the introduction section. as:

Investigation of bonding behavior of AA1050/AA5083 bimetallic laminates by roll bonding technique

Investigation of bonding properties of Al/Cu bimetallic laminates fabricated by the asymmetric roll bonding techniques

Mechanical Properties and Microstructural Evolution of Bimetal 1050/Al2O3/5083 Composites Fabricated by Warm Accumulative Roll Bonding

Significant enhancement of bond strength in the roll bonding process using Pb particles

BONDING PROPERTIES OF Al/Al2O3 BULK COMPOSITES PRODUCED VIA COMBINED STIR CASTING AND ACCUMULATIVE PRESS BONDING

- citation to all of the mentioned references is necessary.

add error bar to tables and curves with point data

-add scale bar to fig. 1-5

also do this for fig. 6

-

6. PLOS authors have the option to publish the peer review history of their article (what does this mean? ). If published, this will include your full peer review and any attached files.

**Do you want your identity to be public for this peer review?** For information about this choice, including consent withdrawal, please see our Privacy Policy .

Reviewer #1: No

Reviewer #2: **Yes: ** Nima Ahmadi

Reviewer #3: No

---

## [Author Response · Author response to Decision Letter 1]

21 Jul 2025

Reviewer #1: The manuscript is well-structured and presents original research with high relevance to engineering blasting applications. The hybrid PCA-PNN approach is effective and validated rigorously. Therefore, I recommend acceptance after minor revisions, primarily for language clarity, Revising References and enhanced explanation of classification thresholds and model selection rationale.

Reply:

1.The language and grammar of the full text have been modified.

2.The references have been sorted out and increased.

3.In the abstract, the classification thresholds and model selection rationale are fully explained.

Reviewer #2: The structure of the paper is not well enough to publish in this journal. Without a proper modification, the paper is not acceptable. Here are some strict and challenging comments for the authors to address:

1. The grammar of the paper needs major revisions.

Reply: The language and grammar of the full text have been modified.

2. The abstract should illustrate the novelty clearly; however, the structure of the abstract is not well.

Reply: The abstract is greatly modified, and the novelty of the application of the model in engineering blasting is explained.

3. The literature review is not good. It is recommended to use these works to enhance the literature review. These works can illustrate the application of the matter in the real world:

Polymer electrolyte membrane fuel cell performance Revolutionized: Artificial intelligence-validated asymmetric flow channels enhance mass transport via hybrid analytical-numerical frameworks

Seismic inversion based on principal component analysis and probabilistic neural network for prediction of porosity from post-stack seismic data

Permanent magnet synchronous motor demagnetization fault diagnosis and localization

Smart Design of Conical Vortex Generators for Heat Transfer Enhancement: A Synergy of CFD and Bio-Inspired Optimization

Reply: The introduction part has been modified, and relevant literature has been added.

4. The paper assumes PCA is suitable for dimensionality reduction without explicitly testing for multicollinearity among the 12 indicators.

Reply: The core purpose of principal component analysis is to transform multiple potentially relevant original variables into a small number of principal components (new variables) through linear combination. These principal components are independent of each other (no correlation), while retaining most of the information of the original data. If the original variables are almost independent of each other (no collinearity), the significance of principal component analysis is very small (because the information of each variable is very unique and cannot be reduced by merging); on the contrary, if the original variables have strong collinearity (more information overlap), principal component analysis can effectively ' extract common information ' and replace multiple original variables with a few principal components. Before doing PCA, correlation analysis was added in Section 3.2. By calculating the correlation coefficient matrix, as shown in table 3, it can be seen that the correlation between variables is strong (the absolute value of many correlation coefficients >0.5), indicating that there is more information overlap, PCA can extract principal components well, and the dimension reduction effect is more obvious.

5.While the authors claim that PCA was used to reduce dimensionality and eliminate correlation among indicators, there is no statistical evidence provided that such correlations were problematic in the first place. A correlation matrix or VIF values would have supported this decision

Reply: In Section 3.2, the correlation analysis of the data is added. Specifically, the data is converted into a correlation matrix. Through the correlation between the indicators, it is shown that the data can be reduced by principal component analysis.

6. The classification of slotting performance into four levels (G1–G4) based on PCA scores appears arbitrary. There is no explanation of how the thresholds (e.g., [-5.0677, 2.5339] for G1) were determined or whether they correspond to meaningful physical differences in slitting performance.

Reply: The PCA score is between [-5.0677,6.0617]. In Section 3.3, it has been shown that when the PCA score is greater than 0, the slitting effect is better, and when the PCA score is less than 0, the slitting effect is poor. Therefore, the PCA score is first divided into two parts. Secondly, the part of [-5.0677,0] is divided into two parts according to the average score, G1 [-5.0677, -2.5339] and G2 [-2.5339, 0]. The part of [0, 6.0617] is divided into two parts according to the average score, G3 [0, 3.0309] and G4 [3.0309, 6.0617]. The slitting effect is divided into four grades because the slitting effect is too rough simply from greater than 0 and less than 0 respectively. For example, when the slitting effect is good, the principal component score of the 22 nd experiment is 0.1063, while the principal component score of the 21nd experiment is 6.0617. The PCA scores of the two are quite different and cannot be simply divided together. Therefore, the slitting effect grade is divided into four grades. The classification of slitting effect here is based on the results obtained from this experiment, and there is no actual physical difference. However, the slitting effect of different parameter slitting tube can be evaluated by the principal component score, and the influence of different slitting tube parameters on the slitting effect can be obtained.

7. The PNN model uses 28 samples with 3 input features (after PCA). While the reported accuracy is high, the small sample size raises concerns about overfitting, especially since some classes (e.g., G1 and G4) contain very few samples.

Reply: Because it is a model experiment, all the data are obtained through experiments. For example, the width of the groove, in the diameter of 8mm round pipe, cutting out the millimeter level gap, it is difficult to cut out a large number of different width of the gap ; it is also difficult to carry out a large number of experiments on the thickness of the slit tube within the limited range of the cartridge diameter of 6mm and the opening of 10mm. Therefore, there are fewer samples of experimental data, but all experiments are true and reliable. The PCA score is used to divide the level of slitting effect. For example, there are fewer samples in G1 and G4, indicating that among different slitting parameters, only a few parameters seriously affect the slitting effect, resulting in excellent or poor slitting effect.

Reviewer #3: interesting work . but some corrections are needed

specify the novelty through the text

lack of related references. the following references are recommended to ad to the introduction section. as:

Investigation of bonding behavior of AA1050/AA5083 bimetallic laminates by roll bonding technique

Investigation of bonding properties of Al/Cu bimetallic laminates fabricated by the asymmetric roll bonding techniques

Mechanical Properties and Microstructural Evolution of Bimetal 1050/Al2O3/5083 Composites Fabricated by Warm Accumulative Roll Bonding

Significant enhancement of bond strength in the roll bonding process using Pb particles

BONDING PROPERTIES OF Al/Al2O3 BULK COMPOSITES PRODUCED VIA COMBINED STIR CASTING AND ACCUMULATIVE PRESS BONDING

citation to all of the mentioned references is necessary.

add error bar to tables and curves with point data

-add scale bar to fig. 1-5

also do this for fig. 6

Reply: Relevant literature has been added in the introduction part. Modify Figure 1 to Figure 6 and increase the scale.

---

## [Decision Letter · Decision Letter 1]

4 Aug 2025

Slotting blasting model experiment and PCA-PNN evaluation model of influencing factors of slitting effect

PONE-D-25-31588R1

Dear Dr. sun,

We’re pleased to inform you that your manuscript has been judged scientifically suitable for publication and will be formally accepted for publication once it meets all outstanding technical requirements.

Kind regards,

Mithilesh K. Dikshit

Academic Editor

PLOS ONE

Additional Editor Comments (optional):

Reviewers' comments:

Reviewer's Responses to Questions

**Comments to the Author**

1. If the authors have adequately addressed your comments raised in a previous round of review and you feel that this manuscript is now acceptable for publication, you may indicate that here to bypass the “Comments to the Author” section, enter your conflict of interest statement in the “Confidential to Editor” section, and submit your "Accept" recommendation.

Reviewer #1: All comments have been addressed

Reviewer #2: All comments have been addressed

Reviewer #3: All comments have been addressed

2. Is the manuscript technically sound, and do the data support the conclusions?

Reviewer #1: Yes

Reviewer #2: Yes

Reviewer #3: Yes

3. Has the statistical analysis been performed appropriately and rigorously? 

Reviewer #1: Yes

Reviewer #2: Yes

Reviewer #3: Yes

4. Have the authors made all data underlying the findings in their manuscript fully available?

Reviewer #1: Yes

Reviewer #2: Yes

Reviewer #3: Yes

5. Is the manuscript presented in an intelligible fashion and written in standard English?

Reviewer #1: Yes

Reviewer #2: Yes

Reviewer #3: Yes

6. Review Comments to the Author

Reviewer #1: (No Response)

Reviewer #2: I have completed my review of your manuscript titled "[Slotting blasting model experiment and PCA-PNN evaluation model of influencing

factors of slitting effect]" and I am pleased to inform you that I recommend ACCEPTANCE of your paper for publication in PLOS ONE.

Reviewer #3: (No Response)

7. PLOS authors have the option to publish the peer review history of their article (what does this mean? ). If published, this will include your full peer review and any attached files.

**Do you want your identity to be public for this peer review?** For information about this choice, including consent withdrawal, please see our Privacy Policy .

Reviewer #1: **Yes: ** Muhammad Azeem Ullah

Reviewer #2: **Yes: ** Nima Ahmadi

Reviewer #3: No

---

## [Editor Report · Acceptance letter]

PONE-D-25-31588R1

PLOS ONE

Dear Dr. Sun,

I'm pleased to inform you that your manuscript has been deemed suitable for publication in PLOS ONE. Congratulations! Your manuscript is now being handed over to our production team.

Kind regards,

on behalf of

Dr. Mithilesh K. Dikshit

Academic Editor

PLOS ONE